# Strengthening contraceptive counselling services to empower clients and meet their needs: protocol for a two-stage, multiphase complex intervention in Pakistan and Nigeria

Nguyen Toan Tran,[1] Moazzam Ali [iD],[2] Syed Khurram Azmat [iD],[3] Armando Seuc,[2] Funmilola M Olaolorun,[4] Muhammad Ali Awan,[3] Imran Morhason-Bello [iD],[5] Ellen Mpangananji Thom,[6] Joseph Martin,[7] Hanifah Datti Abubakar,[8] Qudsia Uzma,[6] James Kiarie [iD] [2]

**Correspondence to**
Dr Moazzam Ali;
alimoa@who.int

## ABSTRACT

**Background** High-quality contraceptive counselling can accelerate global efforts to reduce the unmet need for and suboptimal use of modern contraceptives. This study aims to identify a package of interventions designed to strengthen in Pakistan and Nigeria and determine their effectiveness in increasing client-level decision-making, autonomy and meeting of contraceptive needs.

**Methods** A multisite, two-stage and five-phase intervention design will start with a pre-formative, formative, design, experimental and reflective phase. The pre-formative phase will map potential study sites and establish the sampling frame. The two-part formative phase will first use participatory approaches to identify clients' perspectives, including young couples and providers, to ensure research contextualisation and address each interest group's needs and priorities followed by clinical observations of client–provider encounters to document routine care. The design workshop in the third phase will result in the development of a package of contraceptive counselling interventions. In the fourth experimental phase, a multi-intervention, three-arm, single-blinded, parallel cluster randomised-controlled trial will compare routine care (arm 1) with the contraceptive counselling package (arm 2) and the same package combined with wider methods availability (arm 3). The study aims to enrol a total of 7920 participants. The reflective phase aims to identify implementation barriers and enablers. The outcomes are clients' level of decision-making autonomy and use of modern contraceptives.

**Ethics and dissemination** Ethical approval for this study was obtained from the WHO Scientific and Ethics Review Committee (Protocol ID Pakistan: ERC 006232 and Nigeria ERC: 006523). Each study site is required and has obtained the necessary ethical and regulatory approvals that are required in each specific country. Findings will be presented at local, national and international conferences and disseminated by peer-review publications.

**Trial registration number** NCT06081842.

## STRENGTHS AND LIMITATIONS OF THIS STUDY

⇒ A mixed-method study design offers valuable insights and can address research questions that quantitative or qualitative methods alone cannot adequately answer.
⇒ Multicentre research offers several benefits compared with studies done at a single centre such as access to a larger diverse sample, the ability to pool resources and foster connections and collaborations among researchers.
⇒ The study will be implemented in a limited geography of the target countries; therefore, caution needs to be taken before generalising the findings.

## BACKGROUND

Overlooking women's and couples' need for quality contraceptive counselling constitutes a missed opportunity in family planning programmes and services to ensure equitable access to the health, social and economic benefits of modern contraception for women, men and adolescents. Access to family planning is a human right enshrined in the United Nations' Sustainable Development Goal 3 on health and well-being and, specifically, its target 3.7: *ensure universal access to sexual and reproductive health-care services, including for family planning*.[1] Around 164 million women of reproductive age have an unmet need for modern contraceptive methods despite increased contraceptive use over the past decades.[2] Satisfying the unmet need for contraception in low-income and middle-income countries and offering all pregnant women and their newborns the standard care recommended by the WHO would result in dramatic reductions of unintended

pregnancies by 68%, unsafe abortions by 72%, maternal deaths by 62% and neonatal deaths by 69%.[3] Adequate contraceptive coverage among users can be improved by addressing suboptimal use and high discontinuation rates.[4]

Effective contraceptive counselling can help individuals choose a method that meets their needs and preferences, manage any side-effects and continue using their preferred method or switch to an alternative one.[5] Contraceptive counselling could improve the effective use of modern contraception and reduce unmet need according to a systematic review that synthesised the evidence on the comparative effectiveness of different contraceptive counselling strategies.[6] The quality of contraceptive counselling in low-resource settings is often low. In Senegal, for example, only 18% of providers counselled their clients on the three-item Method Information Index (method use, side-effects and timing of follow-up visits).[7] Interventions targeting women requesting or initiating a chosen method, including structured counselling on side-effects, tend to show positive effects on contraceptive continuation.[8–11] Counselling interventions targeting family planning service users,[12–16] abortion service users,[17–19] postpartum women[20–25] and those reached by community-based interventions[26–28] have inconsistent effects on contraceptive behaviour and satisfaction outcomes. In contrast, increased contraceptive use often follows interventions introducing systematic contraceptive counselling among women attending non-family planning outpatient services.[29–31] However, these interventions often include expanded contraceptive method provision, and the observed changes cannot be attributed to counselling alone. High-quality evidence is generally limited, and no clear consensus exists on how best to deliver contraceptive counselling to meet clients' contraceptive needs and enhance their satisfaction.

Building on this systematic review, the WHO organised a geographically diverse think tank in May 2019 in Geneva, Switzerland, to identify research and guidance gaps and other opportunities to improve the quality of contraceptive counselling.[32] Equally important was the crafting of a definition on contraceptive counselling, which was adopted by think tank members[33]: *Contraceptive counselling is defined as the exchange of information on contraceptive methods based on an assessment of the client's needs, preferences and lifestyle to support decision-making as per the client's intentions. This includes the selection, discontinuation or switching of a contraceptive method. The key principles are based on coercion-free and informed choice; neutral, understandable and evidence-based information; collaborative and confidential decision-making process; ensuring respectful care, dignity and choice.* The definition took into consideration the core dimensions articulated in the literature and mirrors the principles that were deliberated by the think tank: a two-way discussion and decision-making process between the client and provider, sharing objective and user-friendly information, and fundamentally grounded in the rights to autonomy, dignity, privacy, respect and participation,

among others. In addition, person-centeredness is a critical component of quality in family planning.[5] The four-item person-centred contraceptive counselling scale (PCCC) is a valid and reliable measure of person-centred contraceptive counselling that reflects patients' perspectives on contraceptive counselling.[34] For example, integrating Dehlendorf's four-item validated tool: PCCC: person-centred contraceptive counselling scale: Think about your visit. How do you think the member of the healthcare team did? Please rate them on each of the following by circling a number (the scale will use—Poor, Fair, Good, Very good and Excellent). Hence, to measure decision-making autonomy, the following considerations need to factor:

► Respecting me as a person.
► Letting me say what mattered to me about my birth control method.
► Taking my preferences about birth control seriously.
► Giving me enough information to make the best decision about my birth control method.

Furthermore, the think tank members agreed on a prioritised set of guidance, knowledge and implementation research gaps.[32] Such gaps include, for example, understanding or identifying clients' experiences with and expectations of counselling in a variety of global settings; the types of settings different populations, particularly adolescents, unmarried women, migrants, gender nonconforming people and religious/ethnic minorities, receive most of their contraceptive counselling and the quality of counselling across these settings; the fluidity/ambivalence of pregnancy intentions, risk perception (pregnancy and sexually transmitted infections) and women's understanding of their bodies; the independent effect of counselling from method provision; the core elements of high-quality of contraceptive counselling and how these differ by context and what provider characteristics affect counselling quality; the minimum skills and competencies for contraceptive counsellors; the contents of a comprehensive package of best practices for Family Planning (FP) counselling for different populations.

We have reviewed and used the currently available resources noted above such as person-centred care and effective contraceptive counselling approaches, which focus on persons' values and preferences, and are considered to improve the quality of care in family planning. These approaches are likely to promote greater patient autonomy, trust and satisfaction with services. Majority of the research has been based on small, qualitative studies. This study, using a mixed-methods approach, will benefit from existing resources and will also highlight the rights-based approach in the counselling package to understand and document its implementation and effectiveness in the context of two countries.

### Study overall objective and hypothesis
Owing to these gaps combined with the sociocultural challenges related to accessing contraception, we propose a multisite, two-stage and five-phase study with

two main objectives. The objective is to identify and test the effectiveness of a contraceptive counselling intervention bundle on improving clients' informed decision-making autonomy as well as meeting their need for family planning. The outcomes are changes in clients' decision-making autonomy and clients' met need for family planning. The secondary outcome will be a change in modern contraceptive prevalence.

The research rests on the working hypotheses that the package will strengthen existing contraceptive counselling services; incorporate perspectives of key stakeholders such as clients and providers; align with national health policies and WHO global standards and address challenges related to limited public resources.

## Study countries: Pakistan and Nigeria

### Pakistan

The study will be implemented in Pakistan in partnership with the Ministry of National Health Services, Regulations and Coordination. The overall demographic, family planning and reproductive health indicators based on the 2017–2018 Demographic and Health Survey[35] note that rural settings fared worse than urban settings. The total fertility rate was 3.6 children per woman, modern contraceptive prevalence rate was 25%, overall unmet need for family planning was 17% and the demand for family planning satisfied by a modern method was 49%. In terms of family planning knowledge, more than 98% of urban and rural women had heard about any type of modern methods—before marriage, only 15% of women did. Regarding men's attitudes toward contraceptive use, 27% believed that family planning use is a woman's business, while 16% thought it may encourage women to be promiscuous.[36]

### Nigeria

The study will also be implemented in Nigeria by College of Medicine, University of Ibadan. The study will be conducted in two study sites: Lagos (southern part of Nigeria) and Kano (northern part of Nigeria) states. These states were selected for the following reasons: (1) data from the Performance Monitoring for Action programme are available for both states from annual surveys over the past 3 years, including information on the method information index plus and other family planning indicators (pmadata.org); (2) the team has a good network in both locations and will be able to readily collaborate with the State Ministries of Health and (3) safety and security are relatively good with no serious threats in recent times.

The latest demographic, family planning and reproductive health indicators based on the 2018 Demographic and Health Survey in Nigeria in 2019 showed that in general, rural settings fared worse than urban settings.[37] The total fertility rate was 5.3 children per woman; overall unmet need for family planning was 18.9%; demand for family planning satisfied by a modern method was 33.9% and modern contraceptive prevalence rate was 12%. In terms of family planning knowledge, 93.9% of currently married women and 98.3% of sexually active unmarried women had heard about any type of modern contraceptive method.

## METHODS

### Framework and perspectives

#### Theory of change

This health services research is guided by the theory of change framework and incorporates perspectives on human rights and the health system to develop a contraceptive counselling intervention package that strengthens existing family planning services and ensures effectiveness, safety and cultural appropriateness. Following the research findings, the long-term vision is to support the health system in developing a sustainable and scalable contraceptive counselling intervention package for Nigeria and Pakistan. Although the exact package has not yet been developed, based on the requirement set by the respective ministry of health/population in Pakistan and Nigeria, the researchers are willing to contribute to the development of provider training, post-training mentorship and the provision of Information, Education and Communication (IEC) materials to enable the implementation of the intervention bundle.

The theory of change underpins the staged and phased strategy used in this trial, which includes key objectives, activities and intermediate and final results (refer to figure 1).[38]

#### Health system perspective

First and related to the research model outlined in figure 1, the study will rely on the WHO Health System Framework with its six system building blocks and link research and programmatic considerations for contraceptive counselling.[39] The six foundational components of health systems are leadership/governance, healthcare financing, health workforce, medical products and technologies, information and research, and service delivery.[39]

#### Human rights perspective

The research will use the WHO Framework for Ensuring Human Rights in the Provision of Contraceptive Information and Services.[40] The human rights framework comprises nine standards that interweave with the dimensions of quality of care: quality, acceptability, availability, accessibility, non-discrimination, participation, accountability, informed-decision making, and privacy and confidentiality. The research team would ensure a person-oriented approach where no teenage girls or women are discriminated against (eg, in receiving family planning information and services) because of their marital status, age, HIV status, minority status or other social or medical factors.[33]

With regard to the participation standard, the research will ensure active and informed participation of individuals in decision-making that affects them, including on

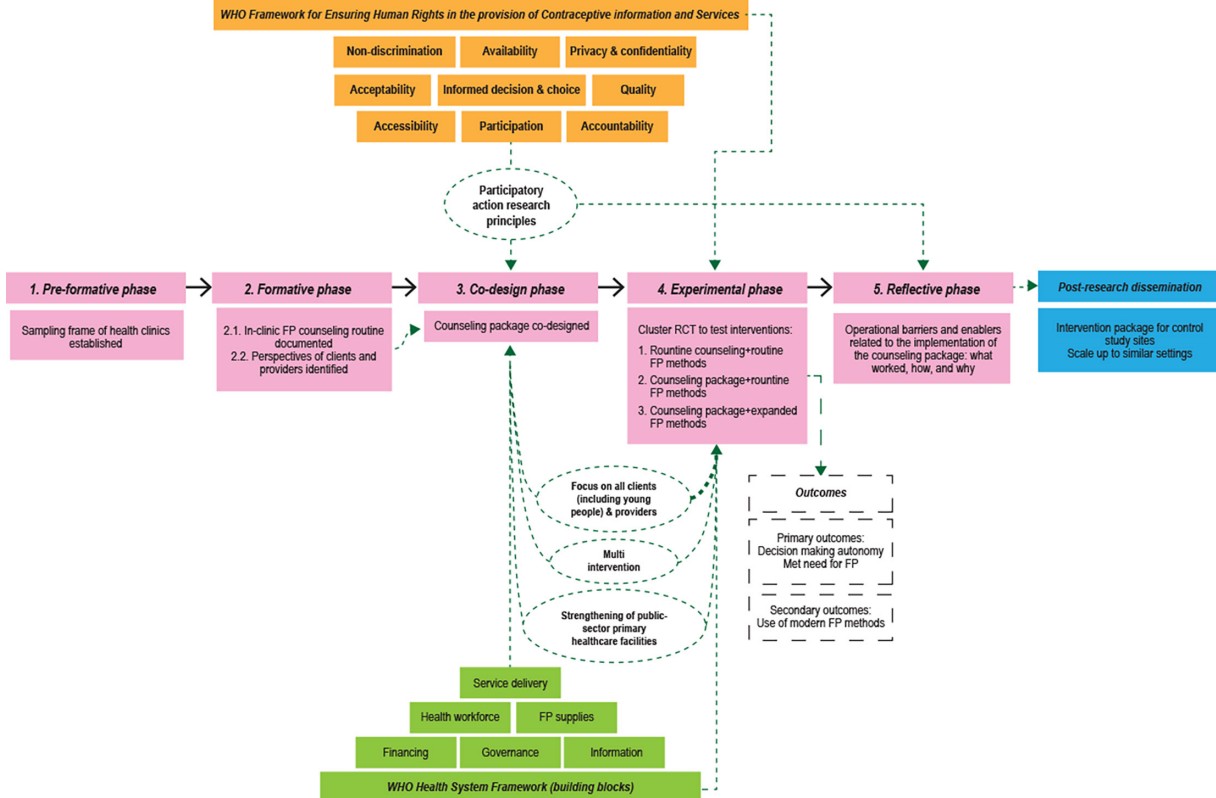

**Figure 1** Overall study model.

matters related to their health.[41] Due to the complex social, cultural and religious challenges and opportunities surrounding contraceptive use, our research design will assume that the following participatory action research (or participatory research) principles will be essential to meet the study objectives as well as gain insights into the implementation process.[42]

### Study phases and timelines
The research entails a complex multisite, two-stage and five-phase (figure 1): Stage 1 is formative and will comprise a pre-formative (phase 1) and a formative (phase 2), while Stage 2 is experimental and will include intervention design (phase 3), experimental (phase 4) and reflective (phase 5). The methods proposed for each phase are not inherently complex; however, the intervention in its globality is complex due to the number and difficulty of behaviours required by both the intervention deliverers and recipients, and the involvement of multiple organisational levels or groups targeted by the intervention, among other factors.[43] The design adheres to recommendations from the for developing a complex intervention, assessing its feasibility and facilitating its implementation and evaluation.[43] As such, the phases are interlinked and have key objectives, methodologies and outcomes.

### Stage 1
The pre-formative phase aims to establish a sampling frame of potential study sites by mapping primary

health centres based on set criteria. Eligibility criteria for health centres include: (1) provision of family planning services; (2) availability of at least three to four modern contraceptive methods, including a barrier method (eg, condom) and short-term and long-term methods (eg, pills, injectables, implants, intra-uterine devices); (3) absence of contraceptive stock-outs over the preceding 3 months in alignment with national policy; (4) a minimum average of 25 family planning encounters per week (or 100 per month) at selected high-performing facilities (both public and private) and at least 12 encounters per week (or 48 per month) at selected low-performing facilities (again, both public and private); (5) non-receipt of family planning technical support (eg, capacity building, supplies) from other organisations at recruitment, with a commitment to refrain from such support during the study and (6) willingness to participate in the study. Based on this sampling frame, four health centres (public and private sector) will be allocated to the formative phase while the others will be randomised to each arm of the experimental phase. The specific objectives of the *formative phase* are to (1) document the in-clinic contraceptive counselling routine using observation checklists and (2) seek through qualitative research the perspectives of clients, including young people, providers and other stakeholders to ensure adequate contextualisation of the research and address needs and priorities of each interest group regarding contraceptive counselling barriers and enablers.

 Tran NT, *et al. BMJ Open* 2024;**14**:e081967. doi:10.1136/bmjopen-2023-081967

*Stage 2*

The specific objective of the *intervention design phase* is to co-design, based on the results of the formative phase and through a participatory and consensus-building approach engaging clients, providers and other main actors, a package of contraceptive counselling to be implemented during the experimental phase. To be integrated into the final package, interventions will need to consider current human resources and clinical capacity, costs, acceptability, feasibility, scalability and other field considerations.

The objective of the *experimental phase* is to assess, using a cluster randomised controlled trial (RCT) design, the impact of the contraceptive counselling bundle on clients' level of decision-making autonomy and demand satisfied for family planning. The three-armed cluster RCT comprises (1) a control arm with contraceptive counselling and care and a range of family planning methods that are available at field level per the national policy and guidelines; (2) an experimental arm implementing the contraceptive counselling package but still with a range of family planning methods that are available (upto four methods) per the national policy and guidelines at field level and (3) another experimental arm with the contraceptive counselling package but with an expanded range of family planning methods (more than four methods) that are recommended by national policy and guidance.

The *reflective phase* objective is to determine operational barriers and catalysts related to the package implementation using qualitative research. Once the package of interventions has been identified—and resources permitting, we might consider conducting a cost-effectiveness analysis to determine the counselling package cost-effectiveness.

The qualitative research and cluster RCT are detailed in the following sections.

## Timelines

The study be tenataively initiated in November 2023 and it will be completed by November 2025. The analysis and report will be drafted by June 2026.

## Qualitative research
### General design

For the qualitative research components of the study during the formative and reflective phases, we will conduct in-depth interviews (IDIs) in Pakistan and Nigeria with various community members (including women, adolescent males and females, partners and influential individuals), health personnel (such as clinicians, managers and community health workers—CHWs), development partners, donor agencies and national policymakers. The formative phase will involve selecting four health centres (both public and private) and their respective coverage areas in each country using a convenient sampling approach. These sites will be chosen from the sampling frame established during the pre-formative phase.

To identify barriers and facilitators related to contraceptive counselling, separate IDIs will be conducted with groups of 6–10 participants from the following categories:

(1) women who have used family planning services at the health centre within the past 12 months: married and unmarried adolescent females (≤19 years), married and unmarried women 19–39 years, married and unmarried women ≥40 years (The study will include Women of Reproductive Age (WRA)—irrespective of their marital status and age group (including migrants refugee people, ethnic and religious minorities, women or men with disabilities, women or men with HIV, etc are inclusive)); (2) married and unmarried men who are husbands or partners of clients who have used family planning services at the clinic within the past 12 months, and married and unmarried adolescent males (<19 years) and (3) family planning service providers.

About the recruitment of men/male participants, the women participants will be first recruited based on their eligibility and recruitment criteria. If the female participant is selected, she will be asked whether she is using the FP method in consensus with her male partner/spouse or using it covertly. If she uses the FP method with the consensus of her male partner/spouse, we will recruit the accompanying male partners/spouses in the study. If the women participant is using the FP method covertly, then the women will be first asked to provide consent to be invited to study participation. If she does not consent, then men/male participants will be identified by the CHWs at the community level. In both cases, formal consent will be taken from the woman whether she is using the FP method covertly or in consensus with her male partner/spouse.

IDIs will also take place with clinic managers, supervisors, CHWs, policymakers and other influential individuals/opinion leaders (selected purposively) who have observed, supervised or monitored family planning services at/or the catchment of the selected public/private health facility. We anticipate that interviews will last between 45 and 60 mins and will take place in a safe, neutral space of the participant's choosing (eg, town hall, office space, on porch outside participant's home, etc). Both the recruiter and the participant will decide and agree on the location of the interview and this will be conveyed to the research team. Alternatively, the team will also identify pay-per-hour spaces as an option to present to the participant. The interview guides will address several key themes pertinent to contraceptive counselling, including: practices, attitudes and knowledge regarding contraception; health systems and sociocultural challenges and enablers affecting contraceptive use; and quality of care, encompassing autonomy, informed decision-making and other human rights considerations in contraceptive services.

These themes will guide the exploration of participants' perspectives and experiences related to contraceptive counselling within the broader context of reproductive health services.

The qualitative research in the *reflective phase* will occur in three health centres in each of the three study arms (nine centres in total). The centres will be purposely sampled to

include the lowest, average and highest performing ones in terms of measured outcomes to enable the broadest possible understanding. Aligned with the theory of change framework, the qualitative research will conduct an implementation evaluation to examine challenges and facilitators associated with the provision of contraceptive counselling and family planning services, assess fidelity to the intervention within experimental groups and explore strategies for potential scalability. IDIs will be conducted separately with service providers, clients across different age groups (≤19 years, 20–39 years, ≥40 years) and clinic managers. The interview guides will address the following topics related to contraceptive counselling: quality of care including autonomy in decision-making and other human rights perspectives, operational challenges and facilitators, and considerations for scaling up the intervention

### Selection criteria and recruitment approach

A standard process of selection and consent-taking will be undertaken at each study site comprising potential study participants for each of the above categories of individuals (including vulnerable populations). A recruitment approach will follow, and the main actors who will facilitate the recruitment process of the potential target respondents for this research are (a) selected service providers, (b) CHWs—working in the catchment area of the selected service providers, (c) community-based opinion leaders, that is, male, and female schoolteachers, social activist/non-governmental organization (NGO) workers. The country Principal Investigator (PI)/Co-PI will train all of them on the study's main objectives, purpose, issues in ethics and the importance of consent.

### Qualitative data management and analysis

At the country level, the study will engage a team of reseachers with expertise in qualitative methods and local dialects.

The interviews will occur in a private location guaranteeing confidentiality. We anticipate that interviews will last between 45 and 60 mins and will take place in a safe, neutral space of the participant's choosing (eg, town hall, office space, on porch outside participant's home, etc). The interviews will be recorded with the participants' informed consent. Research assistants

proficient in transcription will transcribe and translate the audio recordings into English. Accuracy checks will be conducted by comparing the transcripts with the original audio files.

To conduct thematic analysis, qualitative research management software like QSR NVivo V.12 or Atlas.ti will be used. This involves creating a primary codebook to outline all nodes and using it to code the collected data. As analysis progresses, new nodes will be incorporated into the codebook to capture emerging themes and concepts identified during coding. The data will be coded by research assistants skilled in using Nvivo. To ensure coding quality, inter-rater reliability will be assessed by calculating the Cohen's Kappa coefficient, with a threshold of 0.8 or higher considered acceptable for concordance.

During the intervention design workshop, the findings from the formative phase will be shared and discussed with key research team members, clinicans and clients. The aim is to inform the development of a contextually sensitive package of contraceptive counselling interventions from a bottom-up approach. Each workshop participant's perspectives will be taken into account when assessing the potential effectiveness, feasibility, acceptability to women and providers, integration into family planning programmes, sustainability and scalability of potential interventions.

### Cluster RCT

The research will be a multi-intervention, three-arm, single-blinded, parallel, cluster RCT done in selected primary health centres (clusters) in Pakistan and in Nigeria (see figure 2 for the cluster RCT flow diagram, and figure 3 for the schedule of enrolment, interventions and assessments (Standard Protocol Items: Recommendations for Interventional Trials template). Centres will be randomly allocated to the three study arms in matched ratios (1:1:1) on the basis of the number of monthly family planning encounters, the number of available contraceptive types, the ratio of health workers per population in the clinic coverage zone, the location in urban or rural settings and selected district-level variables that may have an influence on the study outcomes, including

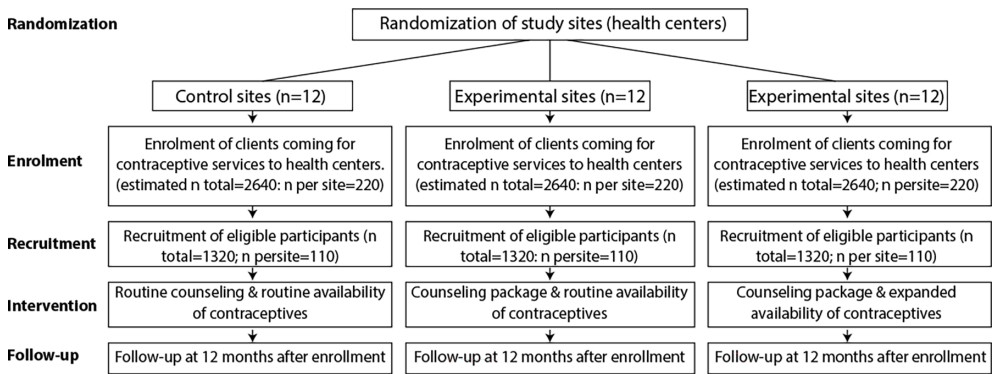

**Figure 2** Flow diagram of the cluster randomised controlled trial in each study country.

| | Enrolment | Allocation | Post-allocation | | | |
|---|---|---|---|---|---|---|
| | $t_{-1}$ | $t_0$ | $t_1$ | $t_2$ | $t_3$ | $t_4$ |
| **TIMEPOINT** | *Initial study family planning visit* | *Initial study family planning visit* | *After initial study visit* | *After each follow-up visit* | *...* | *12 months* |
| **ENROLMENT:** | | | | | | |
| Eligibility screen (estimate) | 7920 | | | | | |
| Informed consent | 3960 | | | | | |
| Allocation | | 3960 | X | X | X | X |
| **INTERVENTIONS & CONTROL:** | | | | | | |
| Counseling package & routine availability of contraceptives | | 1320 | ◆—————————————◆ | | | |
| Counseling package & expanded availability of contraceptives | | 1320 | ◆—————————————◆ | | | |
| Control: routine counseling & routine availability of contraceptives | | 1320 | ◆—————————————◆ | | | |
| **ASSESSMENTS:** | | | | | | |
| **Baseline variables:** Met need for family planning Modern contraceptive use | | X | | | | |
| **Primary outcomes:** Decision-making autonomy Met need for family planning | | | X | X | X | X |
| **Secondary outcomes:** Modern contraceptive use | | | X | X | X | X |

**Figure 3** Schedule of enrolment, interventions and assessments for the cluster randomised controlled trial phase in each study country (SPIRIT template). SPIRIT, Standard Protocol Items: Recommendations for Interventional Trials.

the unmet need for family planning, the level of literacy among women and household income terciles. Only data analysts will be masked/blinded to trial-arm allocation. Service providers skilled in family planning services will implement the clinical components of the contraceptive counselling package.

### Outcome measures

Quantitative indicators will be captured in case-report forms (CRFs) developed and piloted at the country level before finalisation. The proposed outcomes to be assessed after each visit, at mid-point (6 months after enrollment) and at the study end (12 months after enrollment) are oriented toward clients and consist of levels of clients' decision-making autonomy and demand satisfied for family planning.

To determine their level of decision-making autonomy, clients will be asked to answer questions about the different domains contained in the counselling definition (see the Background section). These domains comprise client–provider collaboration, client's needs and preferences, the quality of information, freedom from coercion, confidentiality and dignity. The PCCC, developed and tested in publicly funded clinics in California, will also be used.[43] The four PCCC questions fall under the domains of respect, eliciting—and giving serious consideration to—client's preferences, and information quality and it will be assessed during the intervention phase as per described by UNFPA to measure decision-making autonomy SDG 5.6.1.[44]

The study proposes nonetheless the another outcome, while acknowledging its limitations: modern contraceptive prevalence after the first visit and at 6 and 12 months after enrollment.

### Eligibility criteria for clusters (health centers) and participants

The trial will take place in specific primary health centres that meet the eligibility criteria outlined in the preformative phase objectives. A woman can participate in the study if

1. She comes to the family planning clinic with the intention to (a) use contraception for the first time in her life (new user) or (b) switch from a contraceptive method to another one (switching user) or (c) resume a method after not using any in the prior 3 months (lapse user), or (d) discontinue a modern method (discontinuing user).
2. She is not coming for the resupply of a currently used method, such as pills or injectables.
3. She has the intention to continue her follow-up at the health centre during the 12-month study follow-up.
4. She does not participate in another study.
5. She provides informed consent.

Exclusion criteria: not willing to participate in the complete duration of the trial.

### Interventions

The primary health centres assigned to the first experimental group will implement the intervention package. Meanwhile, those in the second experimental group will implement the intervention package along with an

expanded range of family planning methods recommended by national guidelines. The control group will continue with usual care practices.

Depending on the specific components of the intervention package, certain interventions, such as provider capacity building, may be implemented prior to participant recruitment to ensure readiness and effectiveness of the interventions during the study period.

### Randomisation

In each country, a total of 36 sites will be selected and matched into 12 trios based on the specified criteria. Within each trio, the study sites will be randomly assigned to one of three study arms at a 1:1:1 ratio. This randomisation process will be repeated 12 times, once for each trio. The randomisation process will not have any restrictions, and blinding (masking) of participants and providers cannot be implemented except for data analysts who will handle the study data independently. The study clusters will include all consecutive and eligible participants until the total sample size of 7920 participants is reached.

### Data source and collection

Data pertaining to key outcome measures and quantitative process indicators will be gathered using either paper-based or electronic CRFs. The CRFs will be developed by the WHO team in Geneva in collaboration with the country research teams. To refine the forms, advanced drafts will undergo testing with a suitable sample of service providers and simulated clients from the formative phase sites. Each health centre participating in the RCT will be assigned a research assistant who will receive training on adhering to the study manual and standard operating procedures for data management, developed by the WHO and consistent across both study countries. To ensure data quality, the accuracy and completeness of information recorded on the CRFs will undergo rigorous checks at multiple levels by field coordinators and data managers. This systematic approach will help maintain data integrity throughout the study period.

### Data management

In line with protocols used in other multicenter studies sponsored by the UNDP/UNFPA/UNICEF/WHO/World Bank Special Programme of Research, Development and Research Training in Human Reproduction (HRP) at WHO in Geneva, a web-based data management system with be used for data entry and quality monitoring. HRP's Biostatistics and Data Management team will develop this system using OpenClinica (V.3.16), an electronic data capture software designed for clinical research. Open-Clinica includes programmed edit checks to ensure data quality, completeness and consistency, which helps in reducing delays associated with data queries and problem resolution, ensuring easy accessibility of the dataset. The OpenClinica system is a secure, open-source and reputable web-based software solution that fully adheres to Good Clinical Practice and regulatory guidelines, as well

as the HRP/WHO standard operating procedures for clinical trial management

The country research teams in Pakistan and Nigeria will enter the data locally and have it reviewed by the WHO team in Geneva. We will provide training to the country teams on data entry and management using OpenClinica. The country research teams will be tasked with data verification, uploading data to shared files, and entering it into the OpenClinica system. To enhance accuracy, double data entry will be performed. The system is equipped with automated data query features to identify missing values, outliers, inconsistencies and other errors. The WHO team will oversee data quality monitoring. Any questions regarding data inconsistencies or missing values will be addressed through standardised forms sent to the sites, with continuous resolution of issues as they arise. To ensure data integrity and participant confidentiality, data transmission will be encrypted. Only approved users will be given access to the data, which will be password protected.

### Statistical analysis

We will prepare a comprehensive statistical analysis plan. Participants will form the unit of analysis and the clustered nature of the data will be accounted for. All analyses will be by intention to treat. For continuous variables, we will compute means, SD, minimum and maximum values, and for categorical variables, we will compute frequencies and percentages. We will look at the distribution of variables and see if there are any outliers as part of data quality control and descriptive analysis. Descriptive statistics will be tabulated for individual clusters and aggregated across clusters. Within each of the three groups/arms of the study (two experimental and one control), outcome differences between final (12 months after enrolment) and intermediate (6 months after enrolment) outcome measurements will be obtained, and these changes will be compared between the control group and each of the two experimental groups. For categorical outcomes, $\chi^2$ tests for two independent proportions will be used, and for continuous variables, Student's t-tests for two independent means will be used. We will use IBM SPSS Statistics 25, R V.3.6.2, STATA V.16.1 or SAS V.9.4 statistical packages for the analysis. The findings will be presented using the CONSORT Statement checklist from 2010 and its extension for cluster studies.[45]

### Patient and public involvement

Patients or the public were not involved in the design, or conduct, or reporting, or dissemination plans of this research.

### Ethics and regulatory aspects

We adhered to The Ottawa Statement on the Ethical Design and Conduct of Cluster Randomised Trials when designing this experiment.[46] The Research Project Review Panel (RP2) of the HRP peer-reviewed the protocol, and RP2 and the WHO Research Ethics Review Committee,

Geneva, Switzerland (Pakistan: ERC 006232 and Nigeria ERC: 006523), approved it and so did the IRB of Research and Development Solutions (IRB No:00010843) in Pakistan. While the Ministry of Health, Kano State of Nigeria (NHREC 17/03/2018) and Lagos Research Ethics Committee in Nigeria (LREC/06/10/2180) provided ethical approvals for the Nigeria study.

This research is grounded in human rights principles, specifically participation in the form of participatory action research. The existing literature suggests promise for the success of our proposed approach. For instance, a synthesis of evidence on community-based participatory research across various settings has shown that collaboration among community partners, researchers and organisations has led to community-level actions that improve health and well-being while reducing health disparities.[47] This collaborative process has also contributed to strengthening community capacity in research and leadership skills. Another review highlighted that participatory action research can ensure that research is culturally and logistically appropriate, enhance recruitment capacity, foster professional capacity and competence among stakeholder groups, facilitate productive conflicts followed by constructive negotiation, improve the quality of outputs and outcomes over time, enhance sustainability of project goals beyond funding periods and during gaps in external funding, and drive system changes and new unforeseen projects and activities.[48]

**Author affiliations**
[1]University of Technology Sydney, Sydney, New South Wales, Australia
[2]Department of Reproductive Health and Research, World Health Organization, Geneva, Switzerland
[3]APPNA—Institute of Public Health, Jinnah Sindh Medical University, Karachi, Sindh, Pakistan
[4]Community Medicine, University of Ibadan College of Medicine, Ibadan, Oyo, Nigeria
[5]Department of Obstetrics and Gynaecology, Faculty of Clinical Sciences, College of Medicine/University College Hospital, University of Ibadan, Ibadan, Oyo, Nigeria
[6]World Health Organization Country Office, Islamabad, Pakistan
[7]World Health Organization Country Office, Abuja, Nigeria
[8]Mohammed Abdullahi Wase Teaching Hospital, Nasarawa, Kano, Nigeria

**Acknowledgements** The authors are grateful to Dr Onikepe Owolabi of the Guttmacher Institute and D. Nathalie Roos of the Karolinska Institute for their technical support and insights.

**Contributors** MA, NTT and AS designed the initial study concept, the study protocol and study instruments. SKA, MA, MAA and FMO provided revisions to multiple versions of the study protocol based on the feedback from WHO-ERC. SKA, MA, FMO, MAA, EMT, JM, QU, JK, HDA and IM-B contributed to the final study protocol and study instruments. NTT wrote the first draft of the manuscript with the contributions of MA, AS and JK. While, SKA and MA developed the final draft of the manuscript. All authors read and approved the final manuscript.

**Funding** The project site in Pakistan has been supported by the UNDP-UNICEF-WHO-World Bank Special Programme of Research, Development and Research Training in Human Reproduction Programme (HRP grant No. TRIMS: A66026); a co-sponsored programme executed by the World Health Organisation (WHO). The project in Nigeria generously received funding from United States Agency for International Development (USAID grant No. GHA-G-00-09-00003). The funders will take no role in subjects' recruitment, data collection, analyses, or interpretation, and will take no part in the decision to publish.

**Competing interests** None declared.

**Patient and public involvement** Patients and/or the public were not involved in the design, or conduct, or reporting, or dissemination plans of this research.

**Patient consent for publication** Not applicable.

**Provenance and peer review** Not commissioned; externally peer reviewed.

**ORCID iDs**
Moazzam Ali http://orcid.org/0000-0001-6949-8976
Syed Khurram Azmat http://orcid.org/0000-0003-0158-8734
Imran Morhason-Bello http://orcid.org/0000-0002-7448-4824
James Kiarie http://orcid.org/0000-0003-4180-7858

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
