## [Reviewer comments · BMJ Open]

ARTICLE DETAILS

TITLE (PROVISIONAL)	Strengthening contraceptive counselling services to empower clients and meet their needs: protocol for a 2-stage, multi-phase complex intervention in Pakistan and Nigeria
AUTHORS	Tran, Nguyen Toan; Ali, Moazzam; Azmat, Syed Khurram; Seuc, Armando; Olaolorun, Funmilola; Awan, Muhammad Ali; Morhason-Bello, Imran; Thom, Ellen Mpangananji; Martin, Josephth; Abubakar, Hanifah Datti; Uzma, Qudsia; Kiarie, James

VERSION 1 – REVIEW

REVIEWER	Pearson, Erin University of California San Diego, Medicine
REVIEW RETURNED	14-Dec-2023

GENERAL COMMENTS	1. Strengths and limitations: Many studies use this design (qualitative/participatory to develop and intervention followed by an RCT). Suggest changing from “First research...” to “Uses a qualitative study design...”2. Abstract: Primary outcome(s) should be listed in the Methods.3. Background: Suggest including a discussion of what is known about person-centered care and effective contraceptive counseling approaches (e.g., the BCS+) and how the counseling intervention you develop will build on these.4. Theory of change: In the first paragraph you mention sustainability and scalability as aims, but you also discuss your plan to include multiple interventions, which typically negatively impact both sustainability and scalability. It would be helpful to discuss this further in your Multiple-intervention Strategy section to specify how you will ensure that the package is sustainable and scalable.5. Study phase, Stage 1, eligibility criteria 3: Please specify whether the site can have no stockouts of any of the 3-4 methods they provide, or if you mean that the site cannot have a stockout of all methods (i.e., no methods available).6. Qualitative research: It will be helpful to provide a more thorough description of how men will be recruited. Will they be men who are accompanying their partners for FP services? Men identified by CHWs at the community level? Will their wives/female partners give consent for them to be invited for study participation (would be important in case she is using FP covertly and does not want him to be interviewed).
--

	7. Outcome measures: a. For your outcome “clients’ decision-making autonomy” – are you using two measures? Both the PCCC and an investigator-developed measure of counseling quality? This should be described in greater detail. b. Need to add when the PCCC will be assessed. c. Need to add definition of “met need for FP” – is it modern contraceptive prevalence at each individual timepoint? d. You mention in the Limitations at the beginning of the manuscript that you are only including proximal outcomes. What is the justification for this, especially if you have a 6-month and 12-month follow-up survey? That should enable you to measure more distal outcomes like incident or unintended pregnancy. 8. Interventions: a. It would be helpful to provide information on what you envision to enable implementation of the intervention package (acknowledging that the exact package has not yet been developed). Provider training? Post-training mentorship? Provision of IEC materials? b. Please provide more information on the second experimental group. Will this group be given additional methods to add to their method mix? If so, will training be provided on any methods they did not previously stock (if needed)? Is there a sense of which methods would be added? For example, adding a potentially popular method (e.g., implant) would be very different from adding an unpopular method like female condoms. 9. Randomization: More information is also needed on the third arm here. Your site eligibility criteria just say that they should stock at least 3-4 methods. What about sites that are already stocking 5-6 methods – are they automatically in the third arm? 10. Statistical analysis: No information is provided on how you will assess intervention effectiveness for your primary and secondary outcomes. 11. No information is included on sample size for the cRCT. 12. The manuscript needs a close review to correct typos.
--	---

REVIEWER	Shah, Iqbal Harvard University T H Chan School of Public Health
REVIEW RETURNED	01-Jan-2024

GENERAL COMMENTS	The protocol is well designed and described. Minor points for clarification and some suggestions are as follows:  1. The definition and measurement of primary outcome of "decision-making autonomy" should be elaborated, especially in terms of numerator and denominator of the measures to be estimated. 2. Add definition of terms such as acceptability, feasibility, clinical capacity, scalability, and other terms used for a better understanding. 3. Is the selection criteria outlined in Stage 1 (lines 236-244) would also be used for Stage 2? If so, then the control arm facilities will also have three to four methods available and match on all other criteria as for the two intervention arms. 4. Please clarify if the selection criteria listed in Stage 1 could lead to "selectivity bias" in terms of facilities selected. Criteria 2 of 25
---

	family planning encounters per week or 100 per month may lead to selection of high volume health centers in the two study countries where contraceptive use of modern methods is low. 5. Line 275-276, line 281, line 307: Please clarify how many in-depth interviews per type of respondents. 6. Line 315-316: s the suggested approach to recruit respondents prone to selection bias? 7. Figure 1 is not legible. 8. Please check for minor typos and english, for example in line 81 (change opportunities to opportunity); line 239, add , after implanbts and before intra-uterine device; line 259: remove the second "controlled" before (RCT).
--	---

VERSION 1 – AUTHOR RESPONSE

Reviewer: 1 Dr. Erin Pearson, University of California San Diego Comments to the Author:	Author response/s
1. Strengths and limitations: Many studies use this design (qualitative/participatory to develop and intervention followed by an RCT). Suggest changing from “First research...” to “Uses a qualitative study design...”	The text has been revised as suggested
2. Abstract: Primary outcome(s) should be listed in the Methods.	The text has been updated as per requirements
3. Background: Suggest including a discussion of what is known about person-centered care and effective contraceptive counseling approaches (e.g., the BCS+) and how the counseling intervention you	We have added the details of PCCC and added the following text; We have reviewed and used the currently available resources, such as Person-centered care and effective contraceptive counselling approaches, which focus on persons’ values and preferences and are considered to improve the quality of care in family planning. These approaches will likely promote greater patient autonomy, trust, and service satisfaction. The majority of the research has been based on small, qualitative studies. This study, using a mixed-methods approach, will benefit from

develop will build on these.	existing resources. It has also highlighted the rights-based approach in the counselling package to understand and document its implementation in the context of two countries.
4. Theory of change: In the first paragraph you mention sustainability and scalability as aims, but you also discuss your plan to include multiple interventions, which typically negatively impact both sustainability and scalability. It would be helpful to discuss this further in your Multiple-intervention Strategy section to specify how you will ensure that the package is sustainable and scalable.	Thank you for identifying this. The ethics committee also recommended that the study should focus on a single-intervention strategy rather than multiple. The authors acknowledge the oversight in the manuscript during the submission, and we have now made the edits.
5. Study phase, Stage 1, eligibility criteria 3: Please specify whether the site can have no stockouts of any of the 3-4 methods they provide, or if you mean that the site cannot have a stockout of all methods (i.e., no methods available).	The criteria have been revised and described in detail in the text.
6. Qualitative research: It will be helpful to provide a more thorough description of how men will be recruited. Will they be men who are accompanying their partners for FP services? Men identified by CHWs at the community level? Will their wives/female partners give consent for them to be invited for study participation	We have inserted the following text in the manuscript: "About the recruitment of men/male participants, the women participants will be first recruited based on their eligibility and recruitment criteria. If the female participant is selected, she will be asked whether she is using the FP method in consensus with her male partner/spouse or using it covertly. If she uses the FP method with the consensus of her male partner/spouse, we will recruit the accompanying male partners/spouses in the study. If the women participant is using the FP method covertly, then the women will be first asked to provide consent to be invited to study participation. If she does not consent, then men/male participants will be identified by the CHWs at the community level. In both cases, formal consent will be taken from the woman whether she is using the FP method covertly or in consensus with her male partner/spouse".

(would be important in case she is using FP covertly and does not want him to be interviewed).	
7. Outcome measures:	
a. For your outcome “clients’ decision-making autonomy” – are you using two measures? Both the PCCC and an investigator-developed measure of counseling quality? This should be described in greater detail.	Yes we will use both measures for the outcome “clients’ decision-making autonomy”. The details are expanded in the background section.
b. Need to add when the PCCC will be assessed.	Updated in the text that it will be assessed during the intervention phase as per described by UNFPA to measure decision-making autonomy SDG 5.6.1. We have provided a new reference.
c. Need to add definition of “met need for FP” – is it modern contraceptive prevalence at each individual timepoint?	We have updated the text. It is “demand satisfied” not met need. Percentage of women whose demand is satisfied with a modern method of contraception: The percentage of women (or their partners) who desire either to have no additional children or to postpone the next child and who are currently using a modern contraceptive method. Women using a traditional method are assumed to have an unmet need for modern contraception. Reference: https://www.track20.org/pages/data_analysis/core_indicators/overview.php
d. You mention in the Limitations at the beginning of the manuscript that you are only including proximal outcomes. What is the justification for this, especially if you have a 6-month and 12-month follow-up survey? That should enable you to measure more distal outcomes like incident or unintended pregnancy.	We have updated the text and removed the ‘proximal/distal’ terminology for the outcomes for clarity. “The study will investigate the effect on outcomes i.e., clients' level of decision-making autonomy and use of modern contraceptives. Also the study will be implemented in a limited geography of the target countries; therefore, caution should be taken before generalizing the findings”.
8. Interventions:	

a. It would be helpful to provide information on what you envision to enable implementation of the intervention package (acknowledging that the exact package has not yet been developed). Provider training? Post-training mentorship? Provision of IEC materials?	Updated - Following the research findings, the long-term vision is to support the health system in developing a scalable contraceptive counselling intervention package for Nigeria and Pakistan. Although the exact package has not yet been developed, , the research project plans to contribute to the development of provider training, post-training mentorship and the provision of IEC materials (for second and third arms of intervention) to enable the implementation of the intervention package.
b. Please provide more information on the second experimental group. Will this group be given additional methods to add to their method mix? If so, will training be provided on any methods they did not previously stock (if needed)? Is there a sense of which methods would be added? For example, adding a potentially popular method (e.g., implant) would be very different from adding an unpopular method like female condoms.	Updated the text. Second arm implementing the contraceptive counseling package but still with a range of family planning methods that are available (up to 4 methods) per the national policy and guidelines at field level.
9. Randomization: More information is also needed on the third arm here. Your site eligibility criteria just say that they should stock at least 3-4 methods. What about sites that are already stocking 5-6 methods – are they automatically in the third arm?	Updated the text. The third arm with the contraceptive counseling package but with an expanded range of family planning methods (more than 4 methods) that are recommended by national policy and guidance.
10. Statistical analysis: No information is provided on how you will assess intervention effectiveness for your	Updated the text.

primary and secondary outcomes.	Within each of the three groups/arms of the study (two experimental and one control), outcome differences between final (12 months after enrolment) and intermediate (6 months after enrolment) outcome measurements will be obtained, and these changes will be compared between the control group and each of the two experimental groups. For categorical outcomes, chi-square tests for two independent proportions will be used, and for continuous variables, t-student tests for two independent means will be used.
11. No information is included on sample size for the cRCT.	The relevant information is provided in the text.
12. The manuscript needs a close review to correct typos	Updated and revised the text
Reviewer: 2 Dr. Iqbal Shah, Harvard University T H Chan School of Public Health Comments to the Author:	Author response/s
The protocol is well designed and described. Minor points for clarification and some suggestions are as follows:	Thank you.
1. The definition and measurement of primary outcome of "decision-making autonomy" should be elaborated, especially in terms of numerator and denominator of the measures to be estimated.	Definition: Proportion of women aged 15-49 years (married or in union) who make their own decision on all three selected areas i.e. decide on their own health care; decide on use of contraception; and can say no to sexual intercourse with their husband or partner if they do not want. Only women who provide a "yes" answer to all three components are considered as women who make their own decisions regarding sexual and reproductive health. A union involves a man and a woman regularly cohabiting in a marriagelike relationship. Numerator: Number of married or in union women and girls aged 15-49 years old:

	– for whom decision on health care for themselves is not usually made by the husband/partner or someone else; and – for whom the decision on contraception is not mainly made by the husband/partner; and – who can say no to sex. Only women who satisfy all three empowerment criteria are included in the numerator. Denominator: Total number of women and girls aged 15-49 years old, who are married or in union. Proportion = (Numerator/Denominator) * 100 Source: https://unstats.un.org/sdgs/metadata/files/Metadata-05-06-01.pdf
2. Add definition of terms such as acceptability, feasibility, clinical capacity, scalability, and other terms used for a better understanding.	The study will use the definitions described in this paper: Klaic, M., Kapp, S., Hudson, P. et al. Implementability of healthcare interventions: an overview of reviews and development of a conceptual framework. Implementation Sci 17, 10 (2022). https://doi.org/10.1186/s13012-021-01171-7 Link: https://implementationscience.biomedcentral.com/articles/10.1186/s13012-021-01171-7#citeas
3. Is the selection criteria outlined in Stage 1 (lines 236-244) would also be used for Stage 2? If so, then the control arm facilities will also have three to four methods available and match on all other criteria as for the two intervention arms.	Stage 1 is mainly the formative phase. The criteria for Stage 2, which is an intervention phase, includes the details of inclusion/exclusion criteria, and have been described in the main body.
4. Please clarify if the selection criteria listed in Stage 1 could lead	We have modified the criteria based on your feedback: “have on average at least 25 family planning encounters per week or 100 family planning encounters per month at a selected high-performing facility (each of

to "selectivity bias" in terms of facilities selected. Criteria 2 of 25 family planning encounters per week or 100 per month may lead to selection of high volume health centers in the two study countries where contraceptive use of modern methods is low.	public and private sector) and have on average at least 12 FP encounters per week or 48 FP encounters per months at a selected low-performing facility (each of public and private sector)"
5. Line 275-276, line 281, line 307: Please clarify how many in-depth interviews per type of respondents.	This is already described under the section “general design” in the main body text: “For the identification of contraceptive counselling barriers and enablers, separate IDIs of six to ten (6-10) participants will be conducted for the following categories of individuals (1) women who have used family planning services at the health center in the previous 12 months–: married and unmarried adolescent females (\leq 19 years), married and unmarried women 19-39 years, married and unmarried women \geq 40 years [The study will include Women of Reproductive Age (WRA) - irrespective of their marital status and age group (including migrants refugee people, ethnic and religious minorities, women or men with disabilities, women or men with HIV etc. are inclusive] ; (2) married and unmarried men who are partners or husbands of women who have used family planning services at the health center in the previous 12 months, and married and unmarried adolescent males (<19 years); and (3) family planning service providers”.
6. Line 315-316: s the suggested approach to recruit respondents prone to selection bias?	We have clarified further under the section “general design” in the main body text.
7. Figure 1 is not legible.	Based on feedback, better quality of figure provided.
8. Please check for minor typos and english, for example in line 81 (change opportunities to opportunity); line 239, add , after implanbts and before intra-uterine device; line 259: remove the second "controlled" before (RCT).	Addressed.